

# Deriving Cropland N₂O Emissions from Space-Based NO₂ Observations

Taylor J. Adams[1], Genevieve Plant[1], Eric A. Kort[1]

[1]Department of Climate and Space Sciences and Engineering, University of Michigan, Ann Arbor, MI, USA

*Correspondence to*: Taylor Adams (adamsta@umich.edu) &/or Eric Kort (eakort@umich.edu)

**Abstract:** Croplands are the largest anthropogenic source of nitrous oxide ($N_2O$), a potent greenhouse gas and ozone-depleting substance. Agricultural emissions produce small atmospheric signals with high spatiotemporal variability presenting a large observational challenge. If capable, space-based observations could characterize cropland $N_2O$ emissions from farmlands across the world. No current satellite can resolve near-surface $N_2O$

variations from cropland emissions. However, satellite observations of nitrogen dioxide ($NO_2$), a component of $NO_x$ along with nitric oxide (NO), capture cropland emissions. NO, which quickly converts to $NO_2$ in the atmosphere, and $N_2O$ are co-emitted from soils. Both gases are produced by microbial soil processes, and are emitted in large amounts as a result of excess nitrogen from applied fertilizer. Given their co-emission in croplands, we ask: Can satellite $NO_2$ observations be used to infer $N_2O$ emissions? We examine coincident airborne $N_2O$ and $NO_2$

measurements downwind of California croplands to characterize $N_2O$:$NO_x$ emission relationships from farms. We use these emission ratios to transform estimates of agricultural $NO_x$ emissions derived from space-based TROPOMI $NO_2$ observations to $N_2O$ emissions. We compare these estimates to independent ground and airborne studies in the US Corn Belt and Mississippi River Valley. Space-based estimates are broadly consistent with these ground and airborne studies, suggesting that satellite $NO_2$ observations can be used to infer cropland $N_2O$ emissions. Further

refinement of a $NO_2$ proxy approach for cropland $N_2O$ emissions has the potential to expand observational capabilities to constrain regional and global cropland $N_2O$ emissions and inform process models.

## 1 Introduction:

Nitrous oxide ($N_2O$) is a potent greenhouse gas and ozone-depleting substance with sizable anthropogenic emissions. In a 100-year time frame, $N_2O$ has a global warming potential 298 times that of carbon dioxide

(Butterbach-Bahl et al., 2013; Ravishankara et al., 2009). With a long atmospheric lifetime (over 100 years) and no tropospheric sink, $N_2O$ emitted from the surface travels to the stratosphere where it can react with excited oxygen atoms to produce reactive nitrogen oxide radicals that deplete stratospheric ozone. $N_2O$ is now the largest contributor to stratospheric ozone depletion of actively emitted anthropogenic gases (Ravishankara et al., 2009). Since the 1700's the atmospheric concentration of $N_2O$ has increased by over 60 parts per billion (ppb) (Tian et al.,

2024), with an accelerating rate in recent years (Liang et al., 2022).

Agriculture is a dominant source of $N_2O$ emissions, contributing 3.8 (2.5-5.8) Tg $N_2O$-N per year, or about 22% (15%-34%) of global emissions, from 2007 to 2016, with a 30% increase over the past four decades due to nitrogen fertilization (Tian et al., 2020). This trend is expected to accelerate due to the growing demand for food and resources that support agricultural industries as well as waste and industrial processes, highlighting the urgent need

for mitigation efforts (Davidson and Kanter, 2014).

Much of what we know about agricultural $N_2O$ emissions is the result of near-surface $N_2O$ measurements from soil flux chambers. Observations from chamber systems range from ~10 samples per day (automatic chambers) (Rowlings et al., 2015; Sihi et al., 2020) to as infrequently as daily or monthly scales (manual chambers) (Griffis et al., 2013). The small spatial extent of chamber measurements, along with the availability of coincident auxiliary data

(e.g., soil moisture, N application rates), permits robust mechanistic analyses of soil $N_2O$ emissions. However, given the spatial heterogeneity of $N_2O$ emissions (Lawrence et al., 2021), the small spatial resolution (~1m²) of chamber measurements becomes a limitation when assessing emissions at larger spatial scales. Eddy covariance methods can be used to study $N_2O$ emissions at the field scale, with sensitivity to surface emissions from upwind soils of ~10 to





1000 m² (di Marco et al., 2005). Observational constraints at much larger scales are possible with tall (~100's of meters in height) tower measurements of $N_2O$, which are interpreted with atmospheric transport and inverse modeling to infer $N_2O$ emissions. This provides emissions information integrated over several hundred kilometers at a monthly temporal resolution (Chen et al., 2016; Griffis et al., 2013; Nevison et al., 2018, 2023).

More recently, airborne sampling approaches have demonstrated the potential to bridge the scale gap from flux chambers/eddy-flux to tall-towers. Depending on their flight altitude, airborne measurements can resolve $N_2O$ emissions at the farm (~1-2 km) scale (Gvakharia et al., 2020) while sampling an area of several hundred kilometers (Dacic et al., 2024; Desjardins et al., 2015; Eckl et al., 2021; Gvakharia et al., 2020; Herrera et al., 2021; Jimenez et al., 2005; Kort et al., 2008; Xiang et al., 2013). The limitation of airborne observations is that they capture short time windows and sample targeted regions as part of campaigns such as CalNex (2010) (Xiang et al., 2013), FEAST (2017) (Gvakharia et al., 2020) and MAIZE (2021 & 2022) (Dacic et al., 2024; Kort et al., 2022, 2024a, 2024b).

Current observational methods to constrain $N_2O$ emissions for the study of process-level emission controls or mitigation strategies are currently limited to the targeted ground and airborne approaches detailed above. A remote-sensing, space-based solution would have the potential to assess agricultural $N_2O$ emissions at key spatiotemporal scales and broaden the spatial extent of studies beyond those limited by targeted ground and airborne measurements. However, presently, we cannot directly measure surface-level $N_2O$ signals from cropland emissions with a space-based platform. A promising opportunity, however, lies in the widespread remote sensing of nitrogen dioxide ($NO_2$). Nitric oxide (NO) is co-emitted with $N_2O$ from agricultural soils (Davidson et al., 2000), and the NO is largely oxidized to $NO_2$ within seconds to minutes after emission (Jacob, 1999). Given that $N_2O$ and NO are co-emitted and their emission patterns are driven by similar variables (e.g., fertilizer application) (Harrison et al., 1995; Sanhueza et al., 1990), variability in atmospheric $NO_2$ concentrations from agricultural soils may serve as a useful proxy for corresponding $N_2O$ emissions. Space-based $NO_2$ observations, available globally almost daily from TROPOMI, track spatiotemporal variations in $NO_2$ in both urban (Adams et al., 2023; Goldberg et al., 2019a, b, 2021, 2024; Ialongo et al., 2020; Park et al., 2022) and agricultural (Ghude et al., 2010; Huber et al., 2020, 2024; Lin et al., 2023) areas, and have done so for decades (Gonzalez Abad et al., 2019). Beyond the existence of $NO_2$ instrumentation in space, the relatively short lifetime of $NO_2$ means emissions of $NO_2$ lead to large enhancements concentrated in the boundary layer, providing a large signal to observe. For $N_2O$, in contrast, emissions add only a very small enhancement over large background values (often less than 1ppb signals on background over 330ppb, Dacic et al., 2024), which creates a larger observational challenge.

In this work, we explore the potential of using space-based $NO_2$ observations as a proxy for agricultural $N_2O$ emissions. We first discuss the driving mechanisms for $N_2O$ and NO ($NO_2$) from managed croplands. We then use coincident airborne observations of $NO_2$ and $N_2O$ from the California Research at the Nexus of Air Quality and Climate Change (CalNex) campaign conducted in California in 2010 to derive $N_2O$-to-$NO_x$ emission ratios for large spatial regions commensurate with satellite remote sensing of $NO_2$. The aircraft sampling captures integrated emissions ratios that include heterogeneity of emissions in response to a number of driving process-level variables. We hypothesize that we can apply the observed emission ratio distribution to $NO_x$ emissions derived from satellite $NO_2$ observations to obtain an estimate of $N_2O$ emissions from space-based observations. This then enables observational analyses that cover large regions of the world and can track changes over time. We evaluate this possibility for the corn belt and the Mississippi River Valley in the USA.

## 2 Emissions of $N_2O$ and NO from Managed Croplands

$N_2O$ and NO emissions in agricultural soils result from the microbial processes of nitrification and denitrification, with $N_2O$ predominating during denitrification (Baggs, 2008; Chen et al., 1995; Müller et al., 2003) and NO during nitrification (Skiba et al., 1993). Soil moisture influences these processes, where high water-filled pore space (WFPS) favors denitrification and low WFPS favors nitrification. This moisture dependency contributes to large emissions of $N_2O$ and NO following rainfall events (Kim et al., 2012; Scholes et al., 1997), and poorly drained soils are known to emit more $N_2O$ than well-drained soils, which is an important management consideration for $N_2O$





reduction strategies (Lawrence et al., 2021). Drier soils favor higher NO:$N_2O$ emission ratios, often close to or greater than unity, whereas wet soils can have emission ratios closer to 0 (~0.1) (Anderson and Levine, 1987; Davidson, 1992; Johansson and Sanhueza, 1988; Lipschultz et al., 1981; Tortoso and Hutchinson, 1990). Crop type also influences the NO:$N_2O$ ratio (Anderson and Levine, 1987).

Fertilizer application is the most important common driver of NO and $N_2O$ emissions, and fertilized soils have
higher missions of both trace gases (Harrison et al., 1995; Liu et al., 2017; Sanhueza et al., 1990; Shepherd et al., 1991). The accumulation of fertilizer is also hypothesized to drive large post-rainfall emissions of $N_2O$ (Cardenas et al., 1993; Johansson, 1984; Johansson and Sanhueza, 1988; Levine et al., 1996) $NO_x$ (NO + $NO_2$) (Ghude et al., 2010; Jaeglé et al., 2004; Oikawa et al., 2015; Scholes et al., 1997; Smith et al., 1997).

Given the link between fertilizer application and enhancements of NO and $N_2O$, we hypothesize that spatiotemporal
patterns in $NO_x$ emissions from croplands may be a useful proxy to estimate agricultural $N_2O$ emissions. Over extended spatial and temporal scales that incorporate a variety of soil conditions and crop types, the variability in NO:$N_2O$ emissions should be reduced compared to shorter, more localized observations, such as those made in chamber studies. This integrating effect may increase the fidelity of using emissions ratios to derive $N_2O$ emissions.

### 3 Deriving Emission Ratios from CalNex Airborne $N_2O$ and $NO_2$ Observations

Satellite observations of $NO_2$ from TROPOMI or TEMPO, with ground pixel sizes in the range of 5.5x3.5 - 2x4.75 km, will be sensitive to the integrated emissions that emerge from entire farms and multi-farm conglomerates and counties (~250 km$^2$ (Merlos and Hijmans, 2020). To characterize the cropland emission behavior at comparable spatial scales, we use airborne sampling of $N_2O$ and $NO_2$ to determine an emissions relationship between these gases downwind of agricultural fields. Very few airborne campaigns have been made with continuous, high-accuracy,
high-precision measurements of $N_2O$ and $NO_2$ in agricultural regions. For the analysis here we use observations from one of the few campaigns that collected such measurements, the CalNex campaign in 2010 (Fig. 1A), which sampled the San Joaquin and Sacramento Valleys during 6 flights between May 7 and June 18, 2010 (Data available at: https://csl.noaa.gov/projects/calnex/). In-flight instrumentation included the Harvard/National Center for Atmospheric Research's (NCAR) Dual Quantum Cascade Laser Spectrometer for measurement of $N_2O$ (Jimenez et
al., 2005; Kort et al., 2011), and a chemiluminescence $NO_2$ sensor (Pollack et al., 2010; Ryerson et al., 1999, 2001, 2003). During CalNex, measurements of these gases were reported at a 1s rate, and we applied an additional 5-second centered rolling average to reduce instrument noise. To isolate cropland regions, analysis is restricted to locations >0.04° (~3.7 – 4.4 km) from regions with emissions in the top 1% of the National Emissions Inventory (NEI) (Strum et al., 2017), and to periods when the aircraft was below 500m elevation.

Literature often reports NO:$N_2O$ molecular ratios from chamber studies. In this work we determine $N_2O$:$NO_x$ molecular ratios from the aircraft as this is the factor we apply to satellite-derived $NO_x$ emissions to generate $N_2O$ emissions. Since our observations are not made in near proximity to the soil, we assume all the emitted soil $NO_x$ (primarily NO) has converted in the atmosphere to $NO_2$, consistent with previous studies (Huber et al., 2020; Jacob, 1999). The inverse of our ratios is directly comparable to literature NO:$N_2O$ molecular emissions ratios.

We apply two methods to characterize $N_2O$:$NO_x$ emission relationships from the CalNex airborne dataset. Figure 1B and 1C show histograms of derived emission or enhancement ratios corresponding to each approach overlaid over flight maps with the location of data from those approaches. We use these emission or enhancement ratios to characterize the heterogeneity in the empirical relationship between $N_2O$ and $NO_x$ at the farm to multi-farm scale. Below, we briefly outline each approach.





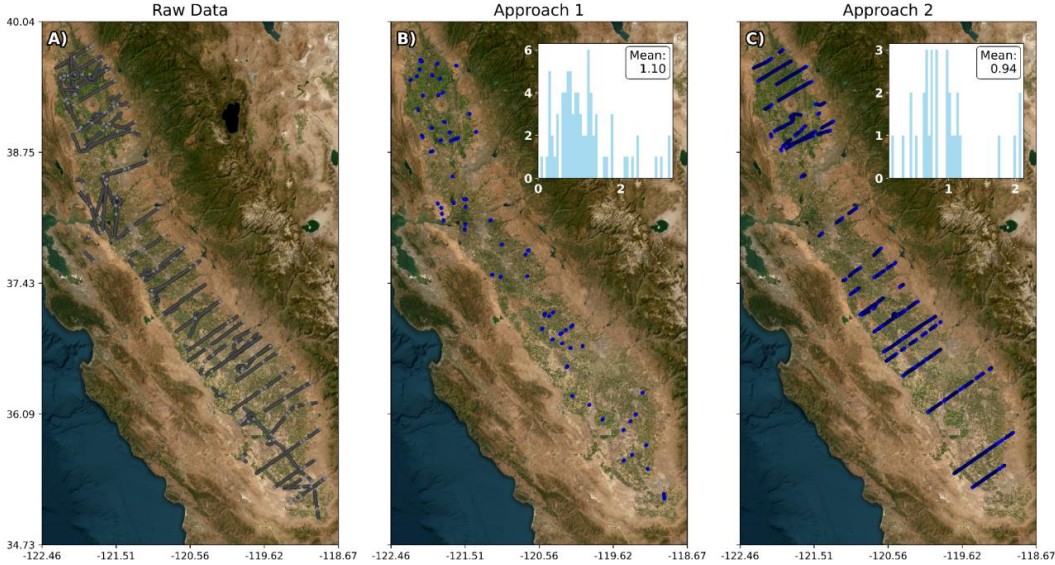


**Figure 1: Flight maps corresponding to data remaining after filtering steps in A) the raw CalNex dataset filtered for data within the agricultural field and away from high NO₂-emission areas, B) in approach 1 and C) in approach 2. A histogram showing the distribution of molecular emission ratios determined in each respective approach is overlaid upon the map. Satellite imagery credit: Esri.**

*Approach 1* seeks to isolate signals from cropland emissions within ~10 km of the aircraft to determine $N_2O:NO_x$ emission relationships. We isolate enhancements from these nearby emissions by filtering for distinct peaks in $N_2O$ concentration and accounting for the chemical loss of $NO_2$. A similar method has been performed to isolate plumes from nearby natural gas flares in the Bakken (Gvakharia et al., 2017). First, we determine $N_2O$ and $NO_2$ enhancements as the concentration above the 5th percentile of a rolling, centered, one-minute window for every data

point. We then isolate cases that are separated by at least 5 seconds in time (~500+ meters in space) where $N_2O$ is enhanced and both $NO_2$ and $N_2O$'s perturbations exceed instrument noise. We require 8 or more observations where the range from minimum to maximum $N_2O$ concentration enhancement exceeds 0.09 ppb, corresponding to greater than $2\sigma$ (0.064 ppb) uncertainty in the mean (based on instrument precision of 0.09 ppb). Cases are removed if their slopes are below zero, as these do not imply co-emission. The distribution of $N_2O:NO_x$ emission ratios post-filtering

cases is shown in Fig. 1B.

   We account for photochemical loss of $NO_2$ by estimating the distance from the source to the observed $N_2O$ enhancement. Assuming the width of an observed enhancement represents a plume's width, we estimate the transport distance from the plume's origin using rural dispersion parameters from Zannetti et al., (2013).We assume a moderately unstable atmosphere during CalNex. Based on daily wind conditions, we estimate the transport time

relative to the average $NO_2$ lifetime and adjust $NO_2$ accordingly to approximate the quantity of emitted $NO_x$. We then derive a $N_2O:NO_x$ emission ratio using type-II ranged major axis regression of the isolated $N_2O$ and $NO_2$ concentration enhancements.

   The average chemistry corrected enhancement emission ratio (referred to as the emission ratio from here forward) is 1.10 ppb $N_2O$ / ppb $NO_x$, a slightly lower value than the ratio (1.22 ppb $N_2O$ / ppb $NO_x$) if chemical loss is not

accounted for. This adjustment is small compared to the variance we see in the ratio. The final dataset using this approach results in 78 individual plumes observed in the nearfield of croplands to derive $N_2O:NO_x$, as shown in Fig. 1B. This approach is designed to isolate enhancements specific to aggregated cropland sources, and the resultant



molecular emissions ratio distribution will encompass variation due to a number of variable drivers of emissions in the upwind region (crop type, fertilizer, soil moisture, etc…).

*Approach 2* incorporates data at broader spatial scales. With this method, we isolate individual flight legs, or portions of the aircraft transects, that are perpendicular to daily wind direction and downwind of agricultural fields. We then treat each flight leg's $N_2O:NO_x$ relationship as a unique enhancement ratio, deriving the background as we do in *approach 1* and assuming impacts from chemical loss are averaged out across flight legs. We do not correct for the chemical loss of $NO_2$, or filter for distinct peaks in $N_2O$ concentration to isolate near-field emissions in this
approach. This approach, rather than isolating small-scale cropland emissions as in approach 1, derives the enhancement relationship across the cropland. This provides significantly more observations but derives an enhancement ratio. To determine the emissions ratio, we then assume chemistry or other processes contribute negligibly, in which case the emissions ratio is equivalent to the enhancement ratio. Approach 2 yields an average molecular emissions ratio of 0.94, determined using type-II ranged major axis regression on the $N_2O$ and $NO_2$
concentration enhancements. Flight legs considered in this method can be visualized in Fig. 1C.

As seen in Fig. 1, we obtain similar mean $N_2O:NO_x$ ratios from each method, demonstrating that the relationship is robust to methodological differences and assumptions. The distributions of the derived $N_2O:NO_x$ values for each approach are also comparable. The Kruskal-Wallis test performed between these two distributions yields a p-value of 0.9983, suggesting no significant difference in median or distribution, and the Welch's t-test, compared between
them, shows there is no significant difference between the emission ratios. Emission ratios range between 0.06 and 3.20 ppb $N_2O$ / ppb $NO_x$ (0.3125 - 16.6 ppb $NO_x:N_2O$) for approach #1 and 0.14 and 2.09 ppb $N_2O$ / ppb $NO_x$ (0.48 - 7.14 ppb $NO_x:N_2O$) for approach #2, reflecting the expected heterogeneity in $N_2O:NO_x$ ratio over agricultural lands, and demonstrating that increasing the spatial scale aggregated to create these ratios dampens variability. These values we observe are in line with literature from soil-chamber measurements which report heterogeneous
$NO_x:N_2O$ emission ratios ranging from near 0 to as high 7 (Johansson and Sanhueza, 1988) in tropical savannahs, or even 10 to 20 in fully aerobic environments (Tortoso and Hutchinson, 1990). This variation occurs as a function of factors such as fertilizer application, crop-type, and other management and environmental factors (Anderson and Levine, 1987; Davidson, 1992; Johansson and Sanhueza, 1988; Lipschultz et al., 1981). As expected with the larger spatial scale of our airborne approach, integrating over variable soil conditions and crop types, our ratios show less
variability than prior soil chamber studies and decrease in variability as more observations are aggregated. In the following sections, we use emission ratios derived from approach #2, though there is little sensitivity to this choice.

**4 Deriving $N_2O$ from Satellite $NO_2$ Observations:**

With $N_2O:NO_x$ emissions ratios derived from the aircraft measurements, $N_2O$ emissions can be determined from space if soil $NO_x$ emissions are calculated from space-based $NO_2$ observations. Many different methods could be
applied to derive $NO_x$ emissions estimates from satellite $NO_2$ observations. For instance, Huber et al., (2020) applied a box model to estimate $NO_x$ emissions from TROPOMI-observed $NO_2$ enhancements within the Mississippi River Valley, Ghude et al., (2010) inferred top-down $NO_x$ emissions from OMI by mass balance, and Lin et al., (2023) estimated soil $NO_x$ emissions in TROPOMI grid cells based on seasonal variation. In principle, the emissions ratios derived in Sect. 3 can be used to estimate agricultural $N_2O$ emissions from space-based observations of $NO_2$. Here,
we use a simple chemical box model and TROPOMI $NO_2$ observations to demonstrate quantification of agricultural $N_2O$ using space-based $NO_2$ observations as a proxy. We focus our analysis on three regions (Figure 2) in the USA where independent ground and airborne measurement campaigns have previously been conducted to determine $N_2O$ emissions, providing a bases for direct comparison with this new approach. This is a robust challenge for this method, as the emissions ratios are determined from aircraft data collected over California in 2010, and crop-type,
management practice, soil moisture, and other driving variables can be quite different in these central US regions.

In this work, we use TROPOMI Version 02.04.00 (S5P_L2__NO$_2$____HiR_2) $NO_2$ retrievals (KNMI). TROPOMI observations are filtered for a quality assurance value greater than or equal to 0.75, indicating high-quality data per the operational retrieval (Van Geffen et al., 2020). We use these TROPOMI $NO_2$ observations in a chemical box





model to determine agricultural $NO_x$ emissions for the Mississippi River Valley, Nebraska, and Iowa, employing a
similar data-model approach as previously outlined in Huber et al., (2020). The chemical box model defined by
equation 1:

$$E_{soil,NO_2} = \frac{U\Delta(NO_{2,VCD})}{X} + \frac{V\Delta(NO_{2,VCD})}{Y} + \frac{V_d(NO_{2,VCD})}{Z_{PBL}} + \frac{NO_{2,VCD}}{t} - E_{NEI},$$ (1)

The first two terms in Eq. 1 captures advection, representing $NO_2$ advected into and out of a domain of interest.
Here, U represents the average zonal wind speed (m/s) across the box of interest and X is the distance of the east-to-
west edge of the domain of interest. $\Delta(NO_{2,VCD})$ is the mean TROPOMI $NO_2$ column enhancement (molecule/m$^2$)
above the background abundance, which we define as the 5th percentile of $NO_2$ abundance in the domain of interest.
V to denote the meridional winds and Y to denote the north-to-south edge distance along the domain of interest. The
third term denotes deposition, where $V_d$ denotes the deposition velocity (m/s) from Yang et al., (2010) (Deposition
velocity by month: DJF, 0.02 c/s; MAM, 0.16 cm/s; JJA, 0.29 cm/s; SON, 0.06 cm/s). $NO_{2,VCD}$ is the average $NO_2$
vertical column density in the domain of interest and $Z_{PBL}$ is the boundary layer height estimate over the domain of
interest. Similar to Huber et al., (2020), $Z_{PBL}$ is set to 1000m over the course of the study. The fourth term represents
chemical loss, where $NO_{2,VCD}$ again denotes the $NO_2$ vertical column density in the domain of interest and "t"
represents the lifetime of $NO_x$. The final term $E_{NEI}$ denotes the average fossil fuel $NO_x$ emissions in the domain of
interest from the 2014 NEI inventory (Strum et al., 2017), averaged monthly to eliminate noise.

In this work, we analyze TROPOMI $NO_2$ retrievals on a daily scale. The size and location of the analysis domain
vary depending upon the region of interest but are at minimum comparable to the size of the box model used in
(Huber et al., 2020) (0.75 x 0.75 degree). We additionally exclude TROPOMI overpasses from the study if they
incorporate less than 30 TROPOMI $NO_2$ observations in our box model domain. Once $NO_x$ emissions are derived
using Equation 1 we multiply soil $NO_x$ emissions ($E_{soil,NOx}$, units of nmole/m2/sec), by the aircraft-derived $N_2O$:$NO_x$
molecular emission ratio to obtain an estimate of $N_2O$ flux, $E_{soil,N2O}$, units of nmole/m2/sec (Equation 2).

$$E_{soil,N_2O} = (ER_{N_2O:NO_x})E_{soil,NO_x,}$$ (2)

To incorporate the impact of variability in the observationally derived emission ratio on estimated $N_2O$ emissions in
this analysis, we employ a Monte Carlo approach where we propagate variation in the $N_2O$:$NO_x$ emission ratio
through to $N_2O$ emissions. We iterate through the daily average values of TROPOMI-derived $NO_x$ emissions and
multiply each by all emission ratios shown in Fig. 1C to derive all possible $N_2O$ emissions from the variation in
$N_2O$:$NO_x$. We then randomly sample one of these $N_2O$ realizations over 10,000 iterations to create a distribution of
TROPOMI $NO_2$-derived $N_2O$ emissions. The 2.5th percentile and the 97.5th percentile define the 95% confidence
interval, with the mean providing a central estimate.

**5 Comparison with Independent Estimates of $N_2O$**

We compare the space-based $N_2O$ emissions estimates with $N_2O$ emissions from independent studies. We first
compare $N_2O$ emissions derived from TROPOMI-$NO_2$ observations with those obtained from chamber
measurements reported by Lawrence et al., (2021). The chamber measurements, conducted between February 2017
and October 2019 in Iowa crop fields, are compared only for the warm season (May-September) of 2018 and 2019
when TROPOMI was operational, and chamber data was available. The comparison domain spans -94.055 to -
93.305 in latitude (0.75 degrees) and 41.605 to 42.355 in longitude (0.75 degrees). The domain of interest lies to the
north of Des Moines, Iowa, and is centered on the Ames, Iowa field site referenced in Lawrence et al.,
(2021)(41.98°N, 93.68°W), and is shown in Fig. 2A, with a star indicating the chamber location.



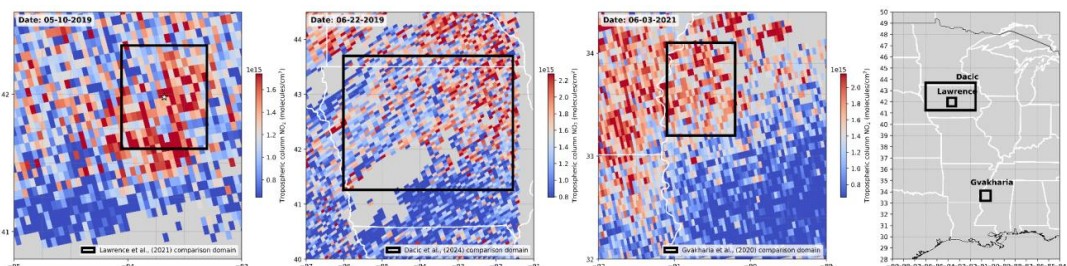

**Figure 2: TROPOMI tropospheric NO₂ columns and the box model domains used for comparison with A) Lawrence et al., 2021 with a star denoting the chamber location, B) Dacic et al., 2024, and C) Gvakharia et al., 2020. D) Each corresponding region shown on a map of the central US.**

We calculate the confidence interval for the Lawrence et al., (2021) chamber dataset by randomly sampling daily average values on days that we also have TROPOMI NO₂-derived N₂O estimates and calculating a 95th confidence interval. For the same observational time-window, we derive daily N₂O emission estimates for the domain of interest using the proxy-method described in Sect. 4. The distribution of chamber-N₂O emissions and its associated 95% confidence interval against the mean and 95% confidence interval for TROPOMI NO₂-derived N₂O flux are shown in Fig. 3A.

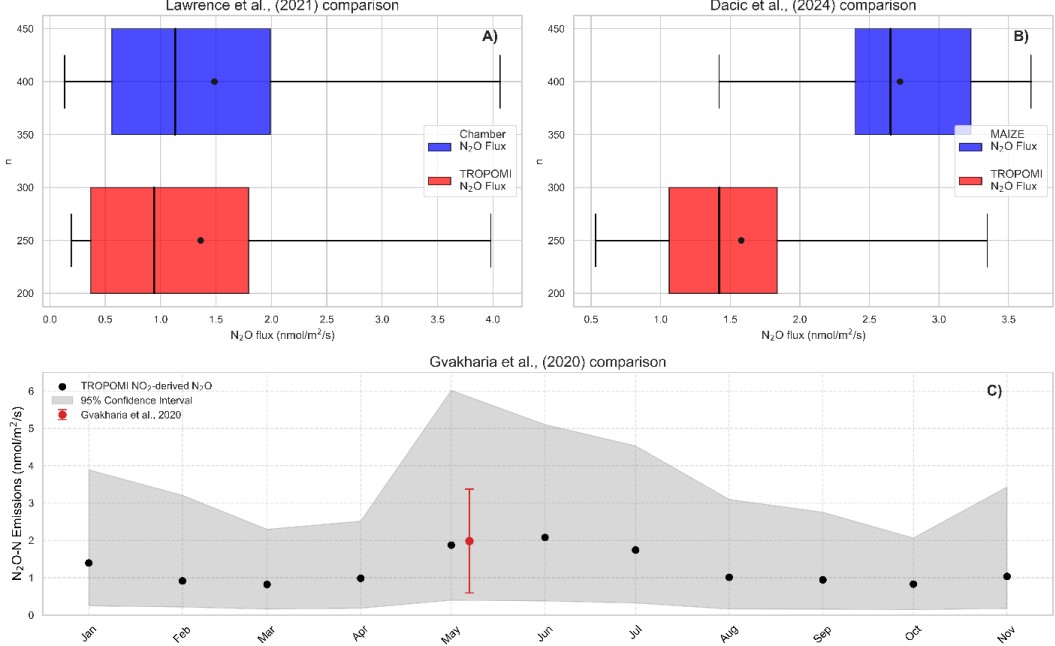

**Figure 3: Satellite derived estimates concurrence with independent studies. (A, top left) Box plots of the distribution of daily average N₂O flux derived from chamber observations detailed in Lawrence et al., 2021 and TROPOMI observations from that period. (B, top right) Box plots of N₂O flux observed by TROPOMI across the MAIZE campaign domain, and ensemble averages for coincident days of the MAIZE (2021-2022) campaign. (C, Bottom) TROPOMI-NO₂ derived N₂O flux (gray distribution) compared against the N₂O flux estimate from Gvakharia et al., (2020) (red band).**

The confidence interval for the mean chamber-derived N₂O flux largely overlaps with that of the TROPOMI NO₂-derived N₂O flux (Fig. 3A). The mean values differ by ~8.7%, with the chamber-derived flux averaging 1.36 (0.561,



5.22; 95% CI) nmol $N_2O/m^2/s$ and the TROPOMI-derived flux averaging 1.49 (0.16, 4.54; 95% CI) nmol $N_2O/m^2/s$. We evaluate the difference in these means with a nonparametric permutation test, which indicated the observed differences between population means were not significantly different. These are independent dataset and methods that operate on different scales. The relative agreement in mean values and the overlap in confidence intervals suggest that the TROPOMI-based approach provides a reasonable estimate of long-term chamber-derived mean $N_2O$ flux in this study.

Next, we compare TROPOMI $NO_2$-derived $N_2O$ fluxes with $N_2O$ emissions derived from aircraft observations during the Measurements of Agriculture Illuminating farm-Zone Emissions of $N_2O$ (MAIZE) 2021 and 2022 campaigns (Dacic et al., 2024; Kort et al., 2022, 2024a, 2024b). These airborne campaigns measured $N_2O$ concentrations over the Iowa croplands during May and June of both years. Observed $N_2O$ concentrations were linked to the surface using an atmospheric transport model, and an ensemble of surface fluxes was derived using a Bayesian inversion framework. In this comparison, we use a larger analysis domain to match the area over which the MAIZE campaigns took place. For 5 flight days of the MAIZE campaign (2021-05-31, 2021-06-03, 2022-05-18, 2022-05-20, and 2022-05-21), we produce a TROPOMI $NO_2$-derived $N_2O$ average flux for the MAIZE domain (S: 40.398; N: 43.399; W: -96.724; E: -90.044, Fig. 2B). We compare this with MAIZE daily ensemble averages. We then randomly sample using the previously described Monte Carlo resampling approach to obtain a 95% confidence interval for MAIZE $N_2O$ flux. Figure 3B shows the comparison between airborne-informed MAIZE $N_2O$ flux and $N_2O$ flux we derive from TROPOMI $NO_2$ observations and our aircraft-derived emission ratios.

While the mean TROPOMI-derived N2O flux is within the 95% confidence interval of the MAIZE emissions, and vice versa (Fig 3B), there is less notable agreement than in the comparison with the chamber emissions from Lawrence et al., (2021) for a similar domain. There exist multiple possible reasons for the larger apparent discrepancy. 1) The comparison is limited to only five days. 2) The MAIZE aircraft flights captured a heavy-tail emissions distribution with small number of fields contributing substantially to total emissions. These high emissions events might be missed by chamber observations, and the coarse scale of the satellite observations used here might reduce sensitivity to small regions with high emissions, thus explaining the TROPOMI and chamber relative agreement with both values lower than determined by flights in MAIZE. 3) It is also possible the enhancement ratio approach used here is failing to capture the emissions ratio as well as desired for this place and time. Still, the mean emissions rate determined from satellite is captured with in the airborne campaign 95% confidence interval, and thus it appears this space-based proxy approach can provide a reasonable mean estimate for this region with five days of observations.

Finally, we compare $N_2O$ fluxes derived using the proposed space-based $NO_2$ proxy method to fluxes derived from airborne mass balance estimates from Gvakharia et al., (2020) over the Mississippi River Valley. During the Fertilizer Emissions Airborne Study (FEAST) in Spring 2017, Gvakharia et al., (2020) observed $N_2O$ fluxes of 1.98 +/- 1.39 nmol $N_2O$-N/$m^2$/s in May 2017, noting significant spatial variation. The TROPOMI instrument was not operational until late 2018, so we do have overlapping data to directly compare TROPOMI $NO_2$-derived $N_2O$ flux estimates with those from Gvakharia et al., (2020). Instead, we calculate emissions for a full calendar year for 2021 and compare these estimates with the May 2017 estimate from the FEAST data (see Fig. 2C and 3C). With the space-based proxy approach, we observe seasonal variation ranging from 0.82 nmol $N_2O$-N/$m^2$/s in March to a peak of 2.1 nmol $N_2O$-N/$m^2$/s in June, and our spring estimate is in close agreement with the estimate from Gvakharia et al., (2020) is shown in Fig. 3C. Specifically, we estimate 1.89 nmol $N_2O$-N/$m^2$/s in May (5.6% difference) using TROPOMI $NO_2$ observations from 2021.

We find the TROPOMI $NO_2$-derived $N_2O$ flux compares favorably with various independent measures of $N_2O$ flux and emissions from the corn belt (Dacic et al., 2024; Lawrence et al., 2021) and Mississippi River Valley (Gvakharia et al., 2020). For two of the comparison, space-based estimates are within ~10% of these independently obtained $N_2O$ estimates, despite this testing capturing multiple regions and time-periods, and the airborne derived emission ratios coming from observations in a completely different agricultural region. This agreement demonstrates the potential of scaling satellite-based $NO_2$ observations with $N_2O:NO_x$ emission ratios to capture agricultural $N_2O$



emissions, and that such an approach may provide a viable method to estimate N$_2$O emissions from agricultural regions around the world.

**6 Conclusion:**

Constraining global emissions of N$_2$O is crucial to understanding its spatiotemporal emission distribution, drivers of emissions, and to guide effective mitigation strategies. Ground-based measurements are inherently limited to the regions where they are installed, while airborne methods are often limited to targeted, short-duration research campaigns. North America has been the focus of many studies, but less is known about the other large agricultural regions around the world. Satellite observations, thus, are an attractive option because they have the potential for more spatial coverage (~global during the warm season) and relatively fine (~daily) temporal resolution. This can also serve to bridge the gap that exists between the present point- and region-scale N$_2$O monitoring. In the absence of direct space-based observations of surface N$_2$O concentrations, we propose to leverage the well-observed trace gas NO$_2$ as a proxy for N$_2$O emissions to create a pathway to monitor N$_2$O emissions in global agricultural regions more comprehensively.

In this work, we derive airborne N$_2$O-to-NO$_x$ emission ratios from the CalNex (2010) airborne campaign around a dense agricultural region of California. These ratios represent the molecular emissions ratio of N$_2$O to NO$_x$ from croplands at spatial scales commensurate with space-based NO$_2$ observations. We combine these ratios with satellite-derived NO$_x$ soil emissions to estimate N$_2$O emissions. We compare our TROPOMI NO$_2$-derived N$_2$O fluxes with N$_2$O emissions estimates from independent chamber observations and two distinct airborne campaigns made in Iowa and the Mississippi River Valley. Our space-based N$_2$O emissions estimates compare favorably with these independent estimates across different regions and as measured by different methods covering different spatial and temporal scales. In comparison with chamber-derived N$_2$O flux (Lawrence et al., 2021), our estimate of mean flux only differs by 8.7%, or ~0.13 nmol N$_2$O/m$^2$/s. Mean estimates differ from airborne emission ratios taken in the Mississippi River Valley (Gvakharia et al., 2020) by ~5.6%. Comparing emissions derived from Bayesian inversion of aircraft data (Dacic et al., 2024) yields a larger N$_2$O flux mean difference by ~50%. In all cases, the confidence intervals of our space-based N$_2$O flux estimates and those from independent measurements and approaches overlap.

In this work, we demonstrate that space-based NO$_2$ observations as a proxy for N$_2$O cropland emissions compare favorably to independent estimates across multiple agricultural areas and years. This suggests that space-based NO$_2$ retrievals are a viable and robust proxy for N$_2$O flux at scales of at least 0.75 x 0.75 degrees, and over timescales as short as five days. Further development and refinement of approaches to characterize agricultural NO$_2$ from satellite observations and link them to N$_2$O emissions are possible. As presented here, the largest source of uncertainty in the estimated N$_2$O emissions derives from the large variability in the observed airborne N$_2$O:NO$_x$ emissions ratio. Improved understanding and definition of this ratio, and what controls variation, could improve the fidelity of this proxy approach. Nonetheless, this work has demonstrated a proxy-based approach that may offer a path towards a more spatially comprehensive constraint on regional and global budgets of agricultural N$_2$O emissions.

**Code Availability**

**Data Availability**

All data used in this manuscript are archived in public databases. Airborne data from P3 aircraft during the CalNex campaign is available from the NOAA Chemical Sciences Laboratory at https://csl.noaa.gov/projects/calnex/. TROPOMI Level 2 Nitrogen Dioxide total column products, Version 02, are available from the European Space Agency at https://doi.org/10.5270/S5P-9bnp8q8. The National Emission Inventory is available from the Environmental Protection Agency at https://www.epa.gov/air-emissions-inventories/2017-national-emissions-inventory-nei-data. Chamber flux measurements from Lawrence et al. (2021) are available at https://doi.org/10.6073/pasta/e6117972f6a80d5f5a9db354957910ed. Airborne data from the FEAST campaign and Gvakharia et al. (2020) is available at https://doi.org/10.7302/Z2XK8CRG. Posterior fluxes from the airborne MAIZE campaign and Dacic et al. (2024) are available at https://doi.org/10.7302/9w5m-mn30.





**Author Contribution**

TA developed the box model code, conducted all model simulations, and designed and optimized the aircraft data
filtering methods. TA also prepared the manuscript, with significant input from EK and GP. EK conceived the
project idea. EK and GP provided guidance on methodology and the presentation of results, and contributed to data
collection and modeling for the experimental datasets used for comparison.

**Competing Interest**

**Acknowledgements**

The authors acknowledge the University of Michigan for supporting this work. The authors thank Dr. Daniel Huber
and Dr. Natasha Dacic for their assistance with running the box model and interpreting the MAIZE results
respectively. The authors also thank the CalNex, FEAST, MAIZE and TROPOMI science teams for their high-
quality data collection and sharing of data into public archives.

**Financial Support**

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
