# Peer review of "Deriving Cropland N2O Emissions from Space-Based NO2 Observations"

_EGUsphere, 2025_

## Author Comment (AC1)

Dear Editor and Reviewers,

We would like to sincerely thank you for your constructive evaluation of our manuscript "Deriving Cropland $N_2O$ Emissions from Space-Based $NO_2$ Observations".

We appreciate your comments and suggestions. We have revised our manuscript thoroughly in response to your comments, and believe the changes strengthen its presentation and scientific contribution.

Some of the most relevant changes we made are:
- Updated our analysis to now include both NO and $NO_2$ when determining $N_2O:NO_x$ emission ratios (rather than determining $N_2O:NO_2$ emission ratios initially)
- Improved the description of our box model to include clarifying details.
- Added a new supplement, including a sensitivity analysis that details how different assumptions about $NO_x$ lifetime and deposition velocity influence results. In the supplement, we now also include $N_2O$ emission estimates and comparison for both our approaches to characterize $N_2O:NO_x$ emissions ratios. We find no significant difference between these approaches in emission ratio distribution or final results.

Below we outline how and where changes were made in the manuscript. We have highlighted all modifications in the revised manuscript using track changes, which includes a couple other editorial corrections/improvements.

Sincerely,
Taylor Adams
On behalf of all authors.

**Reviewer #1:**

**Comment:** In their paper, Adams, Plant and Kort propose to use $NO_2$ space observations to estimate $N_2O$ emissions from farmlands. The paper is well written, has a good set of references and was a pleasure to read. The comparisons with three very different measurement approaches, as shown in Fig 3, is clearly illustrating the potential of the approach. I have a couple of points which I would like to see addressed before the paper is ready to be published.

The method relies on two key steps, first the determination of soil emissions based on satellite columns, and secondly the link between $NO_x$ and $N_2O$ emission fluxes. Several questions came up related to these two steps.

In the paper, $NO_x/N_2O$ emission and $NO_2/N_2O$ concentration ratios are not always clearly distinguished, but emissions are not the same as concentrations and lifetime and $NO/NO_2$ chemical conversion plays a role. It would be good to be more precise at several locations, and describe in more detail how measured concentration ratios are computed back to emission ratios.

*Response: Agreed - we have made changes throughout to be more precise and accurate in discussing NO, $NO_2$, $NO_x$ and emissions, concentrations or enhancements.   Several improvements to the text include:*

1. *Comparison between $NO$:$N_2O$ ratios in the literature to airborne $N_2O$:$NO_x$ ratio derived in this work. Line 123-130: "Literature from chamber studies often reports $NO$:$N_2O$ molecular emission ratios. Chamber measurements of soil emissions typically directly observe NO prior to substantial conversion to $NO_2$. In this work we determine $N_2O$:$NO_x$ molecular ratios from the aircraft, in order to provide a factor that can be multiplied directly to a soil $NO_x$ emissions estimate to generate $N_2O$ emissions.  The soil $NO_x$ estimate could be generated from a range of different approaches, including from space-based $NO_2$ observations as we demonstrate here. While soil emissions of $NO_x$ are predominantly NO, when observed from aircraft or satellite downwind of emissions, the majority of this NO has been converted to $NO_2$ (Goldberg et al., 2024 A; Goldberg et al., 2024 B; Kimbrough et al., 2017; Pilegaard et al., 2013; Seinfeld and Pandis, 2012; Williams et al., 1992)."*

2. *Clarifying the approach for going from a $N_2O$:$NO_x$ enhancement ratio to a corresponding emission ratio to use in our subsequent analysis.*

   *For approach 1, analyzing individual plumes:*
   *Line 147-159: "Approach 1 seeks to isolate concentration enhancement signals from cropland emissions within ~10 km of the aircraft to determine aircraft-derived $N_2O$:$NO_x$ emission relationships. We isolate concentration enhancements from near-field cropland emissions by filtering for distinct peaks in $N_2O$ concentration and accounting for the chemical loss of $NO_2$. A similar method has been performed to isolate plumes from*

*nearby natural gas flares in the Bakken (Gvakharia et al., 2017). First, we determine $N_2O$, NO and $NO_2$ enhancements as the concentration above the 5th percentile of a rolling, centered, one-minute window for every data point from the CalNex airborne NO, $NO_2$ and $N_2O$ dataset. We then isolate cases that are separated by at least 5 seconds in time (~500+ meters in space) where $N_2O$ concentration is enhanced and NO, $NO_2$, and $N_2O$'s concentration perturbations exceed instrument noise. We require 8 or more observations where the range from minimum to maximum $N_2O$ concentration enhancement exceeds 0.09 ppb."*

And line 166-174: *"We then add the observed NO enhancement, thus generating a $NO_x$ enhancement corresponding to emissions from the agricultural field. For each isolated plume, we use the observed $N_2O$ enhancement and chemically corrected $NO_x$ enhancement to calculate a $N_2O:NO_x$ emission ratio using type-II ranged major axis regression. For each plume, the slope represents a unique emission ratio.*

*The average chemistry corrected enhancement emission ratio (referred to as the emission ratio from here forward) is 0.95 ppb $N_2O$ / ppb $NO_x$, a slightly lower value than the ratio (1.36 ppb $N_2O$ / ppb $NO_x$) if chemical loss is not accounted for. This adjustment is small compared to the variance we see in the ratio. The final dataset using this approach results in 76 individual plumes observed in the nearfield of croplands to derive the $N_2O:NO_x$ emission ratio, as shown in Fig. 1B."*

*For approach 2, analyzing individual flight legs:*
*Line 178-186 "We then characterize each flight leg's $N_2O:NO_x$ relationship and treat it as a unique concentration enhancement ratio. Similar to approach 1, we derive the background by defining the enhancement as the concentration greater than the 5th percentile of a rolling centered one-minute window around every data point. This approach, rather than isolating small-scale cropland emissions as in approach 1, derives an integrated concentration enhancement relationship of $N_2O$ to $NO_x$ across a portion of the cropland. We assume impacts from chemical loss are averaged out across flight legs and therefore do not directly correct for the chemical loss of $NO_2$. Therefore, for approach 2, we assume enhancement ratios are equivalent to emission ratios. Approach 2 yields an average molecular emissions ratio of 0.85, determined using type-II ranged major axis regression on the $N_2O$ and $NO_x$ concentration enhancements. "*

**Comment:** l 122: "we assume all the emitted soil $NO_x$ (primarily NO) has converted in the atmosphere to $NO_2$, ..". "The inverse of our ratios is directly comparable to literature NO:$N_2O$ molecular emissions ratios."

Why is this assumption made? This is a potential source of systematic error. Normally the concentration of $NO_2$ is larger than NO, but this depends on the chemical regime, distance from the source and availability of ozone. Also the soil NO/$NO_2$ emission ratio may play a role. Using a chemistry-transport model could lead to more accurate results.

*Response*: *While using $NO_2$ as representative for $NO_x$ is common in satellite based analyses (Goldberg et al., 2024 A; Goldberg et al., 2024 B; Kimbrough et al., 2018; Pilegaard et al., 2013; Seinfelt and Pandis, 2012; Williams et al., 1992), in our analysis of the airborne measurements this is an unnecessary assumption as the aircraft measured both atmospheric concentrations of NO and $NO_2$. We have updated our analysis of the aircraft observations to explicitly include both NO and $NO_2$ in determining $N_2O/NO_x$ emissions ratios, thus explicitly using the full observation of $NO_x$. This makes a very small difference in the end, as almost all the emitted soil NO has converted to $NO_2$ by the time we sample with the aircraft, but now this is explicitly included and is more accurate and precise. Our space-based analysis does rely on the assumption that all emitted soil NO has converted to $NO_2$ when sampled from space. This is a common assumption at the spatial scales of satellite measurements (Goldberg et al., 2024 A; Goldberg et al., 2024 B; Huber et al., 2020; Seinfelt and Pandis, 2012). One could explicitly model this with a full chemical transport model, but this paper is not focused on what might be the optimal approach to quantify soil $NO_x$ emissions from satellite $NO_2$, but rather if empirically derived cropland $N_2O:NO_x$ emission ratios can be used to generate $N_2O$ emission estimates from $NO_x$ satellite data. For this purpose we use a simple box model approach, the results of which we apply the $NO_x$-$N_2O$ proxy approach*

*In the revised manuscript we have updated the relevant mentions of $NO_2$ to $NO_x$.*

**Comment:** l 117: "To isolate cropland regions, analysis is restricted to locations >0.04° (~3.7 – 4.4 km) from regions with emissions in the top 1% of the National Emissions Inventory (NEI) (Strum et al., 2017), and to periods when the aircraft was below 500m elevation."

Pollution from isolated large sources can easily travel long distances (20-100 km). Is this assumption justified and effective in removing non-agricultural contributions? How well are agricultural emissions separated from the other emissions (industry, traffic etc)?

*Response*: *We have taken steps to ensure the impact of large $NO_x$ sources on our analysis is minimized. We have removed data within close proximity to high $NO_x$ emission areas, and account for the influence of large, distant, $NO_x$ sources by defining a local background in the calculation of the enhancement signals used in our analysis. This background is defined as the 5th percentile of $N_2O$ and $NO_x$ within ±30 s, a time-window that corresponds, roughly, to a ~500-700 meter plume (given aircraft velocity). This background approach means enhancements only emerge when they correspond to a plume of this width or narrower. Using the gaussian plume model, a source of this width would be ~2-4 km in distance upwind of the aircraft assuming moderate instability. Sources further than this will impact the background but not the enhancement. In this way, distant (20-100km) pollution sources do not impact our analysis.*

*We have improved the description of the approaches to further clarify this in Section 3: L151-156: "First, we determine $N_2O$, NO and $NO_2$ enhancements as the concentration above the 5th percentile of a rolling, centered, one-minute window for every data point from the CalNex airborne NO, $NO_2$ and $N_2O$ dataset. This background approach means enhancements only*

*emerge when they correspond to a plume of this width or narrower. Using a gaussian plume model, a source of this width would be ~2-4 km in distance upwind of the aircraft assuming moderate instability. Sources further than this will impact the background but not the enhancement. In this way, distant (20-100km) pollution sources do not impact our analysis."*

**Comment:** The explanation of the box model, section 4 equation 1, was confusing, and more discussion (maybe even a figure) could be helpful to increase confidence. The first two terms refer to advection. This would require computing gradients along the wind direction: when downwind concentrations are higher than upwind concentrations this indicates that emissions occur. The authors refer to a delta($NO_2$) as "the mean TROPOMI $NO_2$ column enhancement (molecule/m2) above the background abundance, which we define as the 5th percentile of $NO_2$ abundance in the domain of interest". But this is not the same as a gradient? Please explain more clearly how this is implemented.

***Response****: In the revised manuscript, we have improved our discussion of the  box model to be more descriptive of its design and assumptions. In particular, we clarified that we use the $NO_2$ column enhancement, defined by using a 5th percentile background, as a proxy for a spatial gradient. This assumes that the background value (5th percentile of the data within the domain) is representative of the inflow into the box, and the enhancement is representative of the outflow, which is similar to methods used in previous studies (Godlowska et al., 2023; Huber et al., 2020; Li et al., 2021). The revised discussion is reproduced below:*

*L231-235:"In this work, $\Delta(NO_{2,VCD})$ denotes the mean TROPOMI $NO_2$ column enhancement above a background, here defined as the 5th percentile of $NO_2$ within the box. This enhancement is used as a proxy for the spatial gradient across the domain, where the background value is representative of the inflow into the box and the resulting enhancement is representative of the outflow. This is similar to approaches used in previous studies (Godlowska et al., 2023; Huber et al., 2020; Li et al., 2021)."*

*L (many): throughout the manuscript we have substituted the term "domain of interest" for "box model domain".*

*Citations:*
*Godłowska, Jolanta, et al. "The attempt to estimate annual variability of $NO_x$ emission in Poland using Sentinel-5P/TROPOMI data." Atmospheric Environment 294 (2023): 119482.*
*Huber, D. E., Steiner, A. L., and Kort, E. A.: Daily Cropland Soil $NO_x$ Emissions Identified by TROPOMI and SMAP, Geophys Res Lett, 47, (2020): e2020GL089949.*
*Li, M., et al. "Assessment of updated fuel-based emissions inventories over the contiguous United States using TROPOMI $NO_2$ retrievals." Journal of Geophysical Research: Atmospheres 126.24 (2021): e2021JD035484.*

**Comment:** The authors distinguish deposition and lifetime. How important is the direct deposition term? Normally I expect the reaction with OH to dominate. Please add some more detail on how the lifetime is approximated.

*Response: As you suspect, the deposition term has a much smaller impact than lifetime to reaction with OH. We now include new supplemental figures showing sensitivity studies of these two variables which indeed show this. In supplemental figure 3 & 4, we show the impact on the results using the model as it is described in the main text but with deposition velocity cut in half (Figure S3) and doubled (Figure S4). The impacts, as can be observed, are very small compared to lifetime.*

[Figure]

*Figure S1: Variation of Figure 3, with a slow (7 hour) NO$_x$ lifetime. Satellite derived estimates concurrence with independent studies. (A, top left) Box plots of the distribution of daily average N$_2$O flux derived from chamber observations detailed in Lawrence et al., 2021 and TROPOMI observations from that period. (B, top right) Box plots of N$_2$O flux observed by TROPOMI across the MAIZE campaign domain, and ensemble averages for coincident days of the MAIZE (2021-2022) campaign. (C, Bottom) TROPOMI-NO$_2$ derived N$_2$O flux (gray distribution) compared against the N$_2$O flux estimate from Gvakharia et al., (2020) (red band).*

[Figure]

Figure S2: Variation of Figure 3, with a moderate (5 hour) $NO_x$ lifetime. Satellite derived estimates concurrence with independent studies. (A, top left) Box plots of the distribution of daily average $N_2O$ flux derived from chamber observations detailed in Lawrence et al., 2021 and TROPOMI observations from that period. (B, top right) Box plots of $N_2O$ flux observed by TROPOMI across the MAIZE campaign domain, and ensemble averages for coincident days of the MAIZE (2021-2022) campaign. (C, Bottom) TROPOMI-$NO_2$ derived $N_2O$ flux (gray distribution) compared against the $N_2O$ flux estimate from Gvakharia et al., (2020) (red band).

[Figure]

*Figure S3: Variation of Figure 3, with a slow (half of main text) deposition rates. Satellite derived estimates concurrence with independent studies. (A, top left) Box plots of the distribution of daily average N$_2$O flux derived from chamber observations detailed in Lawrence et al., 2021 and TROPOMI observations from that period. (B, top right) Box plots of N$_2$O flux observed by TROPOMI across the MAIZE campaign domain, and ensemble averages for coincident days of the MAIZE (2021-2022) campaign. (C, Bottom) TROPOMI-NO$_2$ derived N$_2$O flux (gray distribution) compared against the N$_2$O flux estimate from Gvakharia et al., (2020) (red band).*

[Figure]

*Figure S4: Variation of Figure 3, with a fast (double of main text) deposition rates. Satellite derived estimates concurrence with independent studies. (A, top left) Box plots of the distribution of daily average N$_2$O flux derived from chamber observations detailed in Lawrence et al., 2021 and TROPOMI observations from that period. (B, top right) Box plots of N$_2$O flux observed by TROPOMI across the MAIZE campaign domain, and ensemble averages for coincident days of the MAIZE (2021-2022) campaign. (C, Bottom) TROPOMI-NO$_2$ derived N$_2$O flux (gray distribution) compared against the N$_2$O flux estimate from Gvakharia et al., (2020) (red band).*

*Further, in the revised text, we now discuss the source of the lifetime estimates used in this work and the sensitivity of the NO$_x$ emission estimates to the assumption of lifetime.*

*L244-248: "Lifetime values used for the analyses were derived from Martin et al., (2003). In Huber et al., (2020) lifetimes of 3, 5, and 7 hours were used (also derived from Martin et al., (2003)) to constrain the influence of lifetime on the analysis. We opt to use a lifetime value of 3 hours for this main text, but variation in results due to these lifetime values of 5 (moderate) and 7 (slow) hours can be observed in supplemental figures 1 & 2. As was observed in Huber et al., (2020) lifetime has a large impact on NO$_x$ emission estimates."*

**Comment:** TROPOMI is analysed on a daily basis. However, box model emission results based on daily observations may be very noisy. Is noise a problem, especially when comparing campaigns with just a few days of observations. Is this a problem?

*Response: There is noise in the TROPOMI $NO_2$ retrievals; however, we prefer to work at the daily scale to preserve temporal variations in emissions. Aggregating many satellite pixels over a larger domain of interest (as shown in Figure 2), does reduce some of the overall noise on the domain mean. A larger source of 'noise' in our analysis is the availability of satellite data. In our comparisons to independent $N_2O$ studies, we require there be a minimum number of TROPOMI pixels to proceed with the comparison. In the case of the comparison to Dacic et al., this reduces the number of comparison days, which can potentially contribute to a large discrepancy between the two approaches. We have revised the text to include these methodological details.*

L249-254: *"In this work, we analyze TROPOMI $NO_2$ retrievals on a daily scale. The size and location of the analysis domain vary depending upon the region of interest, but have a minimum size 0.75° x 0.75° to be consistent with the domain size used in Huber et al. (2020) . We additionally exclude TROPOMI overpasses from the study if they incorporate less than 30 TROPOMI $NO_2$ observations in our box model domain for the Lawrence et al., (2021) and Gvakharia et al., (2020) comparison, or less than 120 TROPOMI $NO_2$ observations for the larger Dacic et al., (2024) comparison region."*

**Comment:** The emission ratio results are shown in Fig.1. A very broad range of values is observed, from close to 0  to well above 1. This is an important result, and shows that the proposed methodology will not provide good results everywhere. Basically the authors suggest that these differences average out when looking at larger regions. But is this really the case?

*Response: We do not agree with the inference that the observed variability in emissions ratios collected with the aircraft means the proposed methodology will not provide good results everywhere.  These results show that there is heterogeneity in emissions ratios across different crop types, agricultural practices, and environmental conditions (as expected). For our approach, we explicitly include this full variation in estimating $N_2O$ emissions. That is, all this variation is explicitly included and leads to the large confidence intervals we present. Our independent comparison with other regions in the US in different time frames shows remarkably good agreement. This suggests that indeed ratios may be able to be applied elsewhere, likely because these ratios in aggregate converge over larger regions.  More work on observing these ratios from aircraft in a range of environments and conditions would be valuable in understanding the drivers of the spread and potentially provide insight into more refined emissions ratios that could potentially be applied given crop type, soil moisture, etc.*

*We touch on the desirability of future measurements in the conclusion:*

L380-L385: *"Improved understanding and definition of this ratio, and what controls variation, could improve the fidelity of this proxy approach. This could be accomplished with airborne observations of $NO_x$ and $N_2O$.  Capturing different crops, agricultural practices and*

*environmental conditions would provide more insight into emissions ratios and best practices on how to apply to independent satellite data in new domains. This work demonstrates a proxy-based approach that may offer a path towards a more spatially comprehensive constraint on regional and global budgets of agricultural* $N_2O$ *emissions."*

**Comment:** Are $N_2O/NO_2$ ratios expected to be similar in other parts/regions of the world? Can the ratios determined for the San Joaquin valley be used in the other domains in central US (Iowa etc) discussed in Fig.3 ? As mentioned, ratios will depend on moisture, vegetation and soil type, fertilizer use which may vary from one region to another and is also time dependent. This may potentially cause significant biases in the results for a given target region.

*Response: Our distributions of emission ratios (Figure 1B, 1C) generally fall within the emission ratio range from referenced literature. Our largest difference is actually that our dynamic range is dampened relative to literature, which frequently notes these ratios can range to as high as 7 (Johansson and Sanhueza, 1988) or ~10-20 in fully anaerobic environments (Tortoso and Hutchinson, 1990).*

*At this point we can assert that it appears the ratios determined from California can reasonably be applied to these other US regions given the close agreement with independent studies presented here. Extrapolating elsewhere is possible, but would include additional uncertainty, and we have not done so in the analysis here.*

**Comment:** In the conclusion: "As presented here, the largest source of uncertainty in the estimated $N_2O$ emissions derives from the large variability in the observed airborne $N_2O:NO_x$ emissions ratio. Improved understanding and definition of this ratio, and what controls variation, could improve the fidelity of this proxy approach. "

This seems to point towards potential improvements in the methodology. If parameters like soil type, land, moisture and rainfall could be correlated with the emission ratios then this could improve the emission estimates and generalise the method to other regions. Please add some comments at the end of the conclusion how the method may be improved in the future.

*Response: We have expanded this discussion:*

L380-385: "*Improved understanding and definition of this ratio, and what controls variation, could improve the fidelity of this proxy approach. This could be accomplished with airborne observations of* $NO_x$ *and* $N_2O$. *Capturing different crops, agricultural practices and environmental conditions would provide more insight into emissions ratios and best practices on how to apply to independent satellite data in new domains. This work demonstrates a proxy-based approach that may offer a path towards a more spatially comprehensive constraint on regional and global budgets of agricultural* $N_2O$ *emissions."*

**Reviewer #2**:

**Comment:**
This manuscript presents an interesting approach for estimating cropland $N_2O$ emissions using satellite-based $NO_2$ data. The authors attempt to derive $N_2O$ emissions via the $N_2O:NO_x$ ratio. While this is an ambitious effort, the results in their current form are not yet convincing.

From the perspective of nitrogen cycling processes, soil NO and $N_2O$ emissions are governed by different functional genes and enzymes. For instance, soil $N_2O$ emissions are primarily controlled by the $N_2O$ reduction gene (nosZ), whereas NO emissions are regulated by nirK or nirS genes. These genes and enzymes are activated under different soil moisture and oxygen conditions, leading to subsequent gas emissions (Fig.4 in Oswald et al., 2013). After being emitted from the soil to atmosphere, the conversion of NO to $NO_2$ depends on ozone ($O_3$) concentrations and is often incomplete. Therefore, the $NO–NO_2–O_3$ triad is typically studied together in atmospheric chemistry. The authors attempt to estimate column $NO_x$ concentrations from satellite $NO_2$ data and then calculate $N_2O$ concentrations using thespy $N_2O:NO_x$ ratio. This process involves substantial uncertainties that should be carefully quantified at each step.

*__Response__: Soil $N_2O$ emissions are indeed highly heterogeneous, and vary with environmental parameters such as soil moisture. In our study, we aim to include these heterogeneity by characterizing $N_2O:NO_x$ emission ratios from the variety of croplands and conditions sampled during the CalNex airborne campaign. We have made a number of clarifying updates (noted in the response to reviewer 1 and below) to the manuscript and feel that it provides a more comprehensive description of our study.*

*First, some important corrections/clarifications. We do not estimate column $NO_x$ from satellite $NO_2$ and then calculate $N_2O$ concentrations using $N_2O:NO_x$. Some of this confusion may have come about from some imprecise language in our initial draft, which has been improved as explained in the response to reviewer 1. We instead use airborne observations of NO, $NO_2$, and $N_2O$ in order to determine emissions ratios of $N_2O/NO_x$ from operational croplands. We then use satellite observed $NO_2$ columns and a box model to estimate soil $NO_x$ emissions. We then use the determined emissions ratios to convert estimated soil $NO_x$ to soil $N_2O$ emissions estimates, which we compare with independent studies. Our language in the manuscript is now more precise to clearly state this process, which is explicitly considerate of chemistry that is occurring in the atmosphere after emissions of NO from soils.*

*Regarding your comments on ozone concentration and its impact on NO to $NO_2$ conversion, over the course of the CalNex campaign, for all data occurring in or around the agricultural land, the average concentration of O3 was ~44 pppb on 2010-05-11 to ~61 on 2010-06-14. Per theory (Hanrahan et a., 1999), at these concentrations of O3 ~100% of emitted NO should be converted to $NO_2$ in ~200 seconds. In the figure below is the % conversion of $NO_x$ to $NO_2$ by time as is performed in Hanarahan et al., (1999) assuming an average temperature of 20 celsius and a solar zenith angle of 20 degrees.*

[Figure]

*Hanrahan, Patrick L. "The plume volume molar ratio method for determining $NO_2$/$NO_x$ ratios in modeling—Part I: Methodology." Journal of the Air & Waste Management Association 49.11 (1999): 1324-1331.*

*Uncertainty is of course important to consider quantitatively. This manuscript is focused on the question of whether this ratio approach is a viable path to estimate $N_2O$ emissions. As such, our uncertainty analysis focuses on this component, specifically considering the variation in emissions ratio as observed with the aircraft data set with our monte carlo analysis deriving confidence intervals for the estimated $N_2O$ emissions. There is a separate set of uncertainty that is associated with calculating soil $NO_x$ from satellite $NO_2$. We have added a sensitivity study (Supplemental Figures 1-4) to show the impact of this uncertainty on our analysis here. Depending on what approach one chooses to derive soil $NO_x$ from satellite $NO_2$, this uncertainty would change (Section 3).*

*Reference to this new sensitivity analysis:*
*L269-270: "For the box model we use here as an illustrative example, we conduct a sensitivity analysis to lifetime and deposition velocity (model terms 3 & 4), shown in supplementary Fig. 1-4."*

[Figure]

*Figure S1: Variation of Figure 3, with a slow (7 hour) $NO_x$ lifetime. Satellite derived estimates concurrence with independent studies. (A, top left) Box plots of the distribution of daily average $N_2O$ flux derived from chamber observations detailed in Lawrence et al., 2021 and TROPOMI observations from that period. (B, top right) Box plots of $N_2O$ flux observed by TROPOMI across the MAIZE campaign domain, and ensemble averages for coincident days of the MAIZE (2021-2022) campaign. (C, Bottom) TROPOMI-$NO_2$ derived $N_2O$ flux (gray distribution) compared against the $N_2O$ flux estimate from Gvakharia et al., (2020) (red band).*

[Figure]

*Figure S2: Variation of Figure 3, with a moderate (5 hour) $NO_x$ lifetime. Satellite derived estimates concurrence with independent studies. (A, top left) Box plots of the distribution of daily average $N_2O$ flux derived from chamber observations detailed in Lawrence et al., 2021 and TROPOMI observations from that period. (B, top right) Box plots of $N_2O$ flux observed by TROPOMI across the MAIZE campaign domain, and ensemble averages for coincident days of the MAIZE (2021-2022) campaign. (C, Bottom) TROPOMI-$NO_2$ derived $N_2O$ flux (gray distribution) compared against the $N_2O$ flux estimate from Gvakharia et al., (2020) (red band).*

[Figure]

*Figure S3: Variation of Figure 3, with a slow (half of main text) deposition rates. Satellite derived estimates concurrence with independent studies. (A, top left) Box plots of the distribution of daily average* $N_2O$ *flux derived from chamber observations detailed in Lawrence et al., 2021 and TROPOMI observations from that period. (B, top right) Box plots of* $N_2O$ *flux observed by TROPOMI across the MAIZE campaign domain, and ensemble averages for coincident days of the MAIZE (2021-2022) campaign. (C, Bottom) TROPOMI-$NO_2$ derived* $N_2O$ *flux (gray distribution) compared against the* $N_2O$ *flux estimate from Gvakharia et al., (2020) (red band).*

[Figure]

*Figure S4: Variation of Figure 3, with a fast (double of main text) deposition rates. Satellite derived estimates concurrence with independent studies. (A, top left) Box plots of the distribution of daily average N₂O flux derived from chamber observations detailed in Lawrence et al., 2021 and TROPOMI observations from that period. (B, top right) Box plots of N₂O flux observed by TROPOMI across the MAIZE campaign domain, and ensemble averages for coincident days of the MAIZE (2021-2022) campaign. (C, Bottom) TROPOMI-NO₂ derived N₂O flux (gray distribution) compared against the N₂O flux estimate from Gvakharia et al., (2020) (red band).*

**Comment**: The title of the Introduction: remove the colon.

*Response: We have removed this colon.*

**Comment**: L97: and NOₓ?.

*Response: We have added the word "and" to this sentence to improve its clarity.*

**Comment**: Section 2 could be merged with the Introduction for better flow.

*Response: While we agree that section 2 contextualizes this study, we believe this description deserves its own section and provides greater clarity to the reader to remain an independent section.*

**Comment**: *L105–129: How were the $N_2O$ and $NO_x$ emissions validated as originating specifically from cropland? Column concentrations represent a mixture of sources, including direct ground emissions and vertical or horizontal transport from adjacent areas. These concentrations are highly influenced by micrometeorological conditions such as wind speed, wind direction, and surface activities.*

**Response**: *This may be an issue of clarity. Only aircraft observations were used to derive $N_2O$ and $NO_x$ to compute emission ratios.*

*As we noted in a response to reviewer #1, we have removed data within close proximity to high $NO_x$ emission areas. We account for large, distant, anthropogenic $NO_x$ sources to our best ability by defining a local background as the 5th percentile of $N_2O$ and $NO_x$ within ±30 s. This time window corresponds to a ~500-700m plume. Using the gaussian plume model, a source of this width would be ~2-4 km away from the aircraft assuming moderate instability. Thus, more distant sources should perturb the local background over a large enough area that we have removed its $NO_x$, $N_2O$, or $NO$ enhancement.*

*For approach #1, we describe this starting at line 157*

*For approach #2, we have added text to enhance this approach's clarity. This description spans L178-L187.*
*"We then characterize each flight leg's $N_2O$:$NO_x$ relationship and treat it as a unique concentration enhancement ratio. Similar to approach 1, we derive the background by defining the enhancement as the concentration greater than the $5^{th}$ percentile of a rolling centered one-minute window around every data point. This approach, rather than isolating small-scale cropland emissions as in approach 1, derives an integrated concentration enhancement relationship of $N_2O$ to $NO_x$ across a portion of the cropland. We assume impacts from chemical loss are averaged out across flight legs and therefore do not directly correct for the chemical loss of $NO_2$. Therefore, for approach 2, we assume enhancement ratios are equivalent to emission ratios. "*

*To ensure the model's result is clear to future readers, we have added the following text: L242-243: "This model outputs an estimate of non-fossil fuel $NO_x$ emissions for the box model domain.*

**Comment:** L153–154: The value appears quite high and may only be applicable to regions or hotspots with high $NO_x$ and $N_2O$ emissions.

**Response**: *The focus of this study is on managed croplands, we indeed are known for being $N_2O$ and $NO_x$ hotspots. We do not imply these emissions ratios would be representative of non-cropland regions. The emission ratio values may appear high, however, they are broadly consistent with those previously reported in the literature. Our observed emission ratio range is ~0.04 - 4.3 ppb $N_2O$ / ppb $NO_x$ for approach #1, and 0.16 - 1.97 ppb $N_2O$ / ppb $NO_x$ for*

*approach #2. These would translate to ~ 0.2 - ~25 ppb NO / ppb N$_2$O (approach #1) and ~0.5 ~6 ppb NO / ppb N$_2$O as observed by a ground-based chamber. These results are in the range of emission ratios that have previously been observed in literature*

*These values are discussed in Section 2, between lines 87 and 97, as well as in Section 3, between lines 195 and 198.*
*"These values we observe are in line with literature from soil-chamber measurements which report heterogeneous NO:N$_2$O or NO$_x$:N$_2$O emission ratios ranging from near 0 to as high 7 (Johansson and Sanhueza, 1988) in tropical savannahs, or even 10 to 20 in fully aerobic environments (Tortoso and Hutchinson, 1990)"*

**Comment**: The title of Section 3 could be revised to "Materials and Methods."

*Response: We have revised this section's title accordingly.*

**Comment:** L171–186: The current comparisons are not sufficiently convincing. The authors are encouraged to include more site-to-site comparisons—at least five sites, in my view.

*Response: It would be great to have more sites and regions of comparison, though five is an arbitrary number. We actually feel quite fortunate to be able to challenge our approach with three truly independent regions and observational approaches – this is actually a stark challenge for our approach. There are few papers that report N$_2$O emissions after the launch of TROPOMI (the NO$_x$-observing satellite used in this study), which became operational in late-2018. We include comparisons with studies that vary in time, spatial scale, and methodology. If there are specific broader comparisons that were performed after operational TROPOMI observations were made available, with publicly available data that is relevant to cropland N$_2$O emissions, we welcome this broader comparison.*

**Comment:** Units should be expressed as nmol m$^{-2}$ s$^{-1}$.

*Response: We have corrected the formatting of units in Figure 3.*

**Comment:** L279: "N$_2$O" should be formatted as N$_2$O.

*Response: We have corrected this formatting error.*

**Comment:** The uncertainty in the N$_2$O flux calculations should be quantified.

*Response:We include estimates of uncertainty by conducting a monte carlo simulation to demonstrate how variability in N$_2$O:NO$_2$ mixing ratio or daily NO$_x$ emission rates may impact the analysis, and by taking multiple approaches to derive emission ratios - finding little difference in their distribution or results.*

*To further address potential uncertainties impacting the $N_2O$ flux calculation, we have included a sensitivity study shown in supplemental figures demonstrating how different assumptions about $NO_x$ lifetime and deposition velocity would influence the results of this study. These variables are the primary driver of uncertainty in the model, and showing how results vary as we perturb these variables demonstrates our model's sensitivity to them.*

*Given the focus of this manuscript is the viability of the application of ratios to independent soil $NO_x$ emissions estimates, that is where our uncertainty analysis is focused.  Different approaches could be used to derive soil $NO_x$ from satellite $NO_2$, with different uncertainty impacts, but that is not pertinent for evaluating the viability of the emissions ratio approach we investigate here.*

**Comment**: The potential applications of the space-based $N_2O$ flux method should be further discussed.

***Response***: *We have expanded our discussion of potential applications as requested:*

*L380-385: "Improved understanding and definition of this ratio, and what controls variation, could improve the fidelity of this proxy approach. This could be accomplished with airborne observations of $NO_x$ and $N_2O$. Capturing different crops, agricultural practices and environmental conditions would provide more insight into emissions ratios and best practices on how to apply to independent satellite data in new domains.  This work demonstrates a proxy-based approach that may offer a path towards a more spatially comprehensive constraint on regional and global budgets of agricultural $N_2O$ emissions."*